

# New genicular joint angle criteria for flexor muscle (*Musculus Semimembranosus*) during the terrestrial mammals walking

Fumihiro Mizuno[1] and Naoki Kohno[1,2]

[1] Graduate School of Life and Environmental Sciences, University of Tsukuba, Tsukuba, Ibaraki, Japan
[2] National Museum of Nature and Science, Tsukuba, Ibaraki, Japan

## ABSTRACT

**Background:** The genicular or knee joint angles of terrestrial mammals remain constant during the stance phase of walking; however, the angles differ among taxa. The knee joint angle is known to correlate with taxa and body mass among extant mammals, yet several extinct mammals, such as desmostylians, do not have closely related descendants. Furthermore, fossils lose their soft tissues by the time they are unearthed, making body mass estimates difficult. These factors cause significant problems when reconstructing the proper postures of extinct mammals. Terrestrial mammals use potential and kinetic energy for locomotion; particularly, an inverted pendulum mechanism is used for walking. This mechanism requires maintaining the rod length constant, therefore, terrestrial mammals maintain their joint angle in a small range. A muscle reaction referred to as co-contraction is known to increase joint stiffness; both the agonist and antagonist muscles work simultaneously on the same joint at the same time. The *musculus semimembranosus* flexes the knee joint and acts as an antagonist to muscles that extend it.

**Methods:** Twenty-one species of terrestrial mammals were examined to identify the elements that constitute the angle between the *m. semimembranosus* and the tibia based on the period between the hindlimb touching down and taking off from the ground. Measurements were captured from videos in high-speed mode (420 fps), selecting 13 pictures from the first 75% of each video while the animals were walking. The angles between the main force line of the *m. semimembranosus* and the tibia, which were defined as $\theta_{sm-t}$, were measured.

**Results:** The maximum and minimum angles between the *m. semimembranosus* and the tibia ($\theta_{sm-t}$) of the stance instance (SI) were successfully determined for more than 80% of the target animals (17 out of 21 species) during SI-1 to SI-13 within ±10° from the mean. The difference between each successive SI was small and, therefore, the $\theta_{sm-t}$ transition was smooth. According to the results of the total stance differences among the target animals, $\theta_{sm-t}$ was relatively constant during a stance and, therefore, average $\theta_{sm-t}$ ($\theta_{ave}$) can represent each animal. Only Carnivora had a significant difference in the correlation between body mass and $\theta_{ave}$. In addition, there were significant differences in $\theta_{ave}$ between plantigrade and unguligrade locomotion.

**Conclusion:** Our measurements show that $\theta_{ave}$ was 100 ± 10° regardless taxon, body mass, and locomotor mode. Thus, only three points on skeletons need to be measured to determine $\theta_{ave}$. This offers a new approximation approach for

Corresponding author
Fumihiro Mizuno,
f.mizuno.86.09@gmail.com

understanding hindlimb posture that could be applied to the study of the hindlimbs of extinct mammals with no closely related extant descendants.

## INTRODUCTION

Hindlimbs act as propulsive devices for terrestrial locomotion (*Demes et al., 1994*). Common terrestrial behaviors require limbs to support body mass against gravity, which means that terrestrial mammals must resist collapsing joints against gravity, requiring the maintenance of extending joints. Although limbs have the same role in that supporting body mass, joint angles differ between species (*Biewener, 1983*, *2005*; *Inuzuka, 1996*; *Dutto et al., 2006*; *Polly, 2007*; *Fujiwara, 2009*; *Dick & Clemente, 2017*). For example, the angles at the knee joint in Asian elephants is approximately 160° (*Ren et al., 2008*), compared to 137° in chacma baboons (*Patel et al., 2013*), 115° in domestic cats, 124° in lions (*Day & Jayne, 2007*). Thus, the limb joint angle is unique to each species; however, the joints have a wider rotatable range than the angle maintained during standing or walking. This causes problems when reconstructing skeletal specimens into an accurate posture when they were alive. In particular, extinct taxa present a significant challenge when reconstructing postures because the angle when they were alive cannot be observed. For example, desmostylian mammals, which do not have closely related living descendants, have been reconstructed in several different postures even though almost complete skeletons of the same species have been unearthed (*Domning, 2002*; *Inuzuka, Sawamura & Watabe, 2006*; *Fujiwara, 2009*). Furthermore, earlier diverging cetaceans, such as pakicetids and ambulocetids, had functional hindlimbs, and extant cetaceans had completely lost their hindlimbs (*Thewissen, Madar & Hussain, 1998*; *Gingerich, 2001*; *Thewissen et al., 2001*; *Madar, 2007*; *Gingerich et al., 2009*, *2017*). In such cases, there are no extant mammals that can be used as references for skeletal reconstruction. Therefore, knowledge of hindlimb postures in terrestrial mammals is important to understand the transition of locomotive ability through mammalian evolution, including the adaptation from land to sea.

Several studies have explored the relationship between limb posture and variables such as taxa, body mass, and skeletal morphology in extant mammals (*Biewener, 1983*, *1989*, *1990*, *2005*; *Day & Jayne, 2007*; *Fujiwara, 2009*; *Fujiwara & Hutchinson, 2012*: *Dick & Clemente, 2017*). These studies indicate that the larger the size of the mammal species the more upright limb posture the species has. However, there are several exceptions to the relationship between limb posture and body mass (*Fujiwara, 2009*). Furthermore, there is a significant problem with estimating the body mass of extinct mammals because fossils have already lost soft tissues by the time they are unearthed. To resolve these problems, it is important to identify joint angle criteria that are unaffected by other factors as possible.

Quadrupedal mammals use potential and kinetic energy to accelerate their center of mass during running and walking (*Cavagna, Heglund & Taylor, 1977*; *Alexander & Jayes,*

*1978*; *Hildebrand, 1984*; *Hildebrand & Hurley, 1985*; *Alexander, 1991*; *Griffin, Main & Farley, 2004*), employing an inverted pendulum movement to walk. This movement allows the quadrupedal mammals to generate the necessary energy to lift and accelerate the center of mass and maintain a constant stride length (*Cavagna, Heglund & Taylor, 1977*; *Griffin, Main & Farley, 2004*). The inverted pendulum requires that the distance between the ground and the center of mass is constant; therefore, the limb joints are maintained within limited range while walking (*Manter, 1938*; *Gray, 1944*; *Goslow, Reinking & Stuart, 1973*; *Goslow et al., 1981*; *Alexander & Jayes, 1983*; *Inuzuka, 1996*; *Fischer et al., 2002*; *McGowan, Baudinette & Biewener, 2005*). When a joint angle is locked against the force of gravity, not only the agonist muscle but also the antagonist muscle work together. This action increases joint stiffness in humans (*Olmstead et al., 1986*; *Louie & Mote, 1987*; *Nielsen et al., 1994*; *Riemann & Lephart, 2002*; *Knarr, Zeni & Higginson, 2012*). Some electromyographic studies of quadrupedal mammals have shown that both agonist and antagonist muscles act simultaneously during the stance phase—the period in which the foot under consideration is in contact with the floor—when the limb supports the body mass (*Engberg & Lundberg, 1969*; *Tokuriki, 1973*; *Deban, Schilling & Carrier, 2012*; *Araújo et al., 2016*). The knee joint maintains an angle owing to extension against gravity, and the *musculus semimembranosus* acts as the knee joint flexor muscle, which is the antagonist muscle of the *m. quadriceps femoris* when the joint extends.

The *m. semimembranosus* attaches to the ischial tuberosity and interior proximal end of the tibia (Fig. 1) (*Böhmer et al., 2020*). These attachment positions do not move, and the involved parts of the skeleton do not change their shape greatly among taxa. Thus, the positional relationship between the muscle and these parts of the skeleton also shows the relationship among skeleton elements. In addition, the angles of the pelvic girdle differ among different body masses (*Polly, 2007*). Therefore, the angle between the line of action of the *m. semimembranosus* and the tibia has a smaller difference than the angle between the femur and tibia among different body masses. Here, we aimed to (1) reveal the joint angle of terrestrial mammals between the *m. semimembranosus* and tibia during walking, (2) explore the relationships between this angle and taxa, body mass, and locomotor mode, and (3) evaluate whether this angle might be suitable as one of the criteria for the reconstruction of hindlimb postures.

## MATERIALS AND METHODS

The angles between *m. semimembranosus* and the tibia *in vivo* were collected from 21 extant species from 21 genera and 14 families within seven orders (Table 1). These species were selected to cover the superorder and order of mammals (Afrotheria, Proboscidea; Euarchontoglires, Primates, Rodentia; Laurasiatheria, Artiodactyla, Carnivora, Perissodactyla; and Marsupialia, Diprotodontia), a wide range of body masses (*i.e.*, from 0.7 kg for *Suricata suricatta* to 4,060 kg for *Elephas maximus*), and three locomotor modes (plantigrade, digitigrade, and unguligrade) (Table 1). All the target animals were kept in zoos at Higashi Park Zoological Gardens (Okazaki, Japan), Higashiyama Zoo and Botanical Garden (Aichi, Japan), Hitachi Kaminé Zoo (Ibaraki, Japan), Toyohashi Zoo and Botanical Park (Aichi, Japan), and Ueno Zoological Gardens (Tokyo, Japan), and all

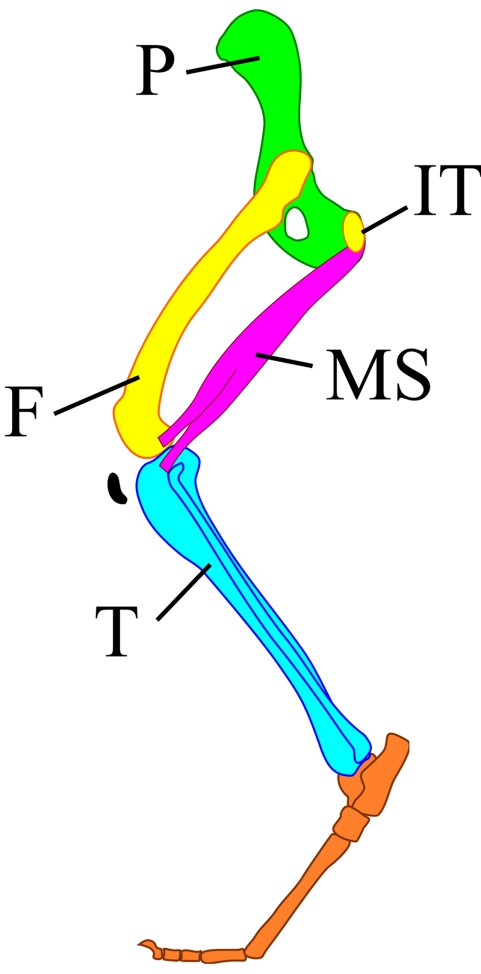

**Figure 1 A model of the mammalian hindlimb bones with *m. semimembranosus* from the exterior view.** *M. semimembranosus* origin locates on the ischial tuberosity (IT) and insertion locates on the interior-proximal end of the tibia (T). The insertion of this muscle was drawn on the exterior of tibia in this figure to identify the position easily, however, the actual insertion locates on the interior of tibia. Abbreviations: F, femur; IT, ischial tuberosity; MS, *m. semimembranosus;* P, pelvis; T, tibia.

observations of living individuals were conducted after gaining official permissions. No significant pathologies or malformations were detected in any of the studied specimens.

All target animals were subjected to video recording using a digital movie camera (EX-FH20; Casio, Tokyo, Japan) in high-speed mode (420 fps). The camera was mounted on a tripod along the visitor viewing route. Therefore, the distance from each target depended on the exhibition/cage arrangement. All videos were taken from the lateral side and at nearly the same level as the target animal when they walked vertically and completely (without stopping, turning, or changing speed) with the camera on flat ground. We waited until each target walked across the camera voluntarily, without any coaxing, meaning it took several weeks or months to obtain the required video footage.

**Table 1 A list of the target animals of this study.**

| Super order | Order | Family | Genus | Species | Locomotor mode | Locality | Body mass (kg) | Body mass reference |
|---|---|---|---|---|---|---|---|---|
| Afrotheria | Proboscidea | Elephantidae | *Elephas* | *maximus* | D | UZ | 4,060.0 | *Shoshani & Eisenberg (1982)* |
| Euarchonto-glires | Primates | Ceropithecidae | *Cercopithecus* | *neglectus* | P | UZ | 4.5 | *Fa & Purvis (1997)* |
| | | | *Chlorocebus* | *aethiops* | P | HKZ | 5.78 | Our study |
| | | | *Macaca* | *fuscata* | P | HKZ | 16.0 | *Obara et al. (2000)* |
| | Rodentia | Cavidae | *Dolichotis* | *patagonum* | D | HP | 8.0 | *Campos, Tognelli & Ojeda (2001)* |
| Laurasia-theria | Artiodactyla | Bovidae | *Ammotragus* | *lervia* | U | UZ | 72.6 | *Cassinello (1997)* |
| | | | *Capra* | *hircus* | U | HKZ | 37.0 | Our study |
| | | Cervidae | *Cervus* | *nippon* | U | UZ | 75.4 | Our study |
| | | | *Rangifer* | *tarandus* | U | HZ | 120.0 | *Garland (1983)* |
| | | Giraffidae | *Giraffa* | *camelopardalis* | U | UZ | 1,000.0 | *Garland (1983)* |
| | Carnivora | Canidae | *Canis* | *lupus* | D | HZ | 43.3 | *Mech (1974)* |
| | | | *Chrysocyon* | *brachyurus* | D | UZ | 23.0 | Our study |
| | | Felidae | *Felis* | *catus* | D | Co | 4.8 | Our study |
| | | | *Panthera* | *leo* | D | TZ | 188.0 | *Haas, Hayssen & Krausman (2005)* |
| | | Herpestidae | *Suricata* | *suricatta* | D | HP | 0.7 | *van Staaden (1994)* |
| | | Ursidae | *Helarctos* | *malayanus* | P | TZ | 45.0 | *Fitzgerald & Krausman (2002)* |
| | | | *Ursus* | *thibetanus* | P | HKZ | 100.0 | Our study |
| | Perissodactyla | Equidae | *Equus* | *cabullus* | U | TZ | 200.0 | *Garland (1983)* |
| | | Rhinocerotidae | *Diceros* | *bicornis* | U | UZ | 1,100.0 | *Hillman-Smith & Groves (1994)* |
| | | Tapiridae | *Tapirus* | *terrestris* | U | HZ | 700.0 | *Padilla & Dowler (1994)* |
| Marspialia | Diprotodontia | Macropodidae | *Macropus* | *giganteus* | O | UZ | 55.0 | Our study |

Notes:
The body mass data are the average of the data from reference or each zoo. Ambulatory style abbreviations: U, unguligrade; D, digitigrade; P, plantigrade; O, other. Institutional abbreviations: HKZ, Hitachi Kamine Zoo, Ibaraki; HP, Higashi Park Zoological Gardens, Aichi; HZ, Higashiyama Zoo and Botanical Garden, Aichi; TZ, Toyohashi Zoo and Botanical Park, Aichi; UZ, Ueno Zoo, Tokyo. Institutions are sorted by alphabetical order.

We selected three videos of each target species that walked with one complete cycle (touching down to the next touching down), straight, and vertical to the camera. Each video was then converted into still images of every frame during the period between touching down and taking off using the GOM Player (GOM & Company, Seoul, South Korea). This period did not depend on time but on the target's behavior. The convert images from the last 25% of each batch (for each measurement period) were discarded because "the muscles that are anatomically positioned to produce limb retraction—the *gluteus superficialis* and *medius*, *semimembranosus* and cranial *biceps femoris*—were active in the second half of swing and approximately the first 50–75% of stance" (*Deban, Schilling & Carrier, 2012*). Subsequently, the first 75% of the stance phase of each step for every specimen was divided into 12 equal time periods (particularly for each step) to obtain 13 images, including the first and last frames (Fig. 2A). The following lines were then drawn

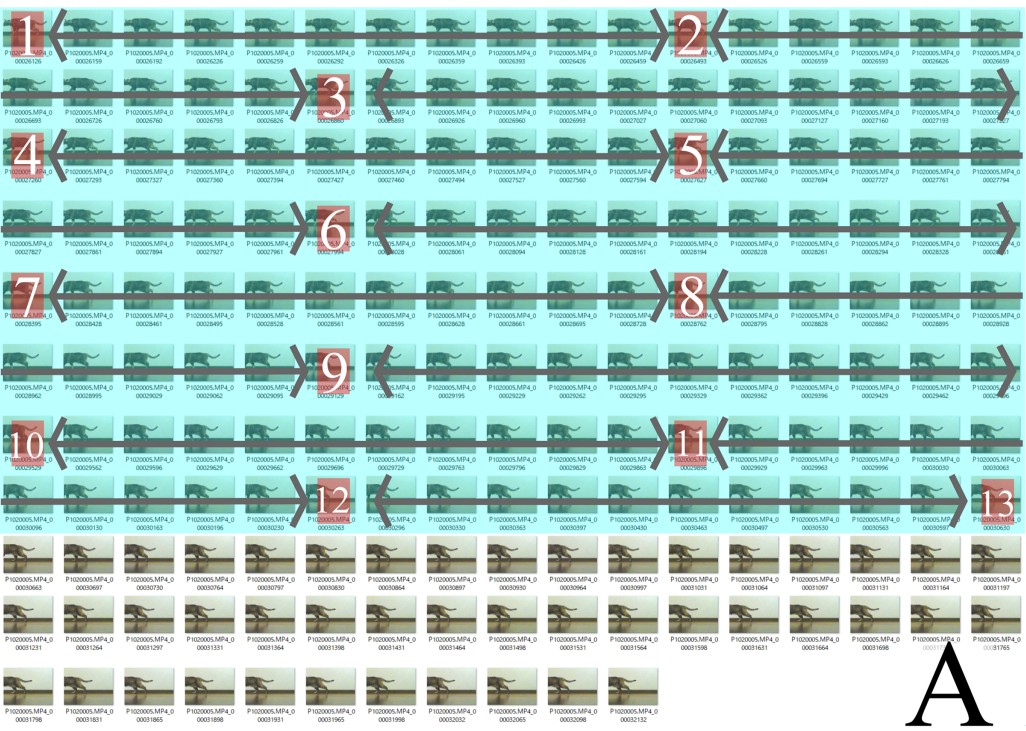

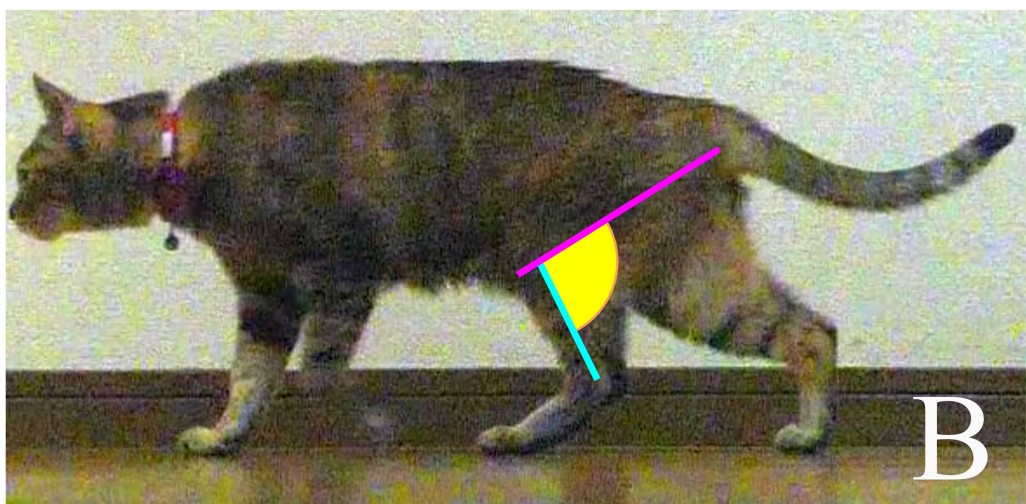

**Figure 2 (A) How the measured pictures were picked.** (B) The line of measured angle on walking cat. (A) A period between touched down to the next touched down was converted to still images. The pictures covered in blue were the first 75% of the total still images. Calculated from the number of first 75% to obtain 13 pictures to measure from the first and the last combined at 12 equal intervals. The numbers in each box are measured pictures. The calculation for intervals is: (number of first 75% of still images)/(13-1) = Interval. (B) A pink line represents a line of action of m. semimembranosus. A blue line represents line of axis of tibia. A yellow area is an angle where is measured.

on each of the 13 pictures using Inkscape (Inkscape project) and the angle between them was measured: a line between the ankle joint and the proximal end of the tibia parallel to the Achilles tendon, and a line between the ischial tuberosity and the proximal end of the tibia (Fig. 2B).

We defined each image of the 13 pictures as a "stance instance" (SI) and numbered them as SI-1 to SI-13. The combination of these 13 images defined a series of a single stance. We measured the joint angle between the lines in each of the 13 images, in each stance, and three stances for each target species in this way. The average angle of each SI was defined as $\theta_{sm-t}$. The body mass of each species was obtained from the literature (Table 1) or zoo records. We compared the transition of $\theta_{sm-t}$ in a stance among species and locomotor modes (unguligrade, digitigrade, and plantigrade), and the average $\theta_{sm-t}$ values (*i.e.*, $\theta_{ave}$) against body mass. Statistical analyses were performed using the R software package (*R Core Team, 2020*). We calculated the standard deviation (SD) to compare the variance of $\theta_{sm-t}$ among taxa, SIs, and locomotor modes. We also calculated correlation coefficient ($r$) to examine relationships between body mass, and $\theta_{ave}$ and performed analysis of variance (ANOVA) to clarify the relationships of $\theta_{ave}$ with body mass, taxa and locomotor modes. For the comparison between $\theta_{ave}$ and body mass, the studied species were divided into the following groups: <1, <10, <100, <1,000, and ≥1,000 kg. The orders and locomotor modes used for the analyses were the same as in the tables. In addition, data that had only one taxon were eliminated, specifically *Elephas*, *Dolichotis*, and *Macropus* in the analysis between taxa; *Macropus* in the locomotor mode comparison; and *Suricata* in the body mass comparison (Table 1).

## RESULTS

Six taxa, *i.e.*, *Elephas* (Proboscidea), *Cervus* and *Rangifer* (Artiodactyla), *Tapirus* (Perissodactyla), and *Felis* and *Panthera* (Carnivora) had differences of less than 10° between the maximum and minimum angles during a stance, which means that $\theta_{sm-t}$ changed within ±5° from the middle. Of the species, *Cervus* had the smallest difference during a stance, 5.80° (±2.9° from the middle-value). Twelve taxa, *i.e.*, *Chlorocebus* and *Macaca* (Primates), *Dolichotis* (Rodentia), *Ammotragus*, *Capra*, and *Giraffa* (Artiodactyla), *Canis*, *Chrysocyon*, *Suricata* and *Helarctos* (Carnivora), and *Equus* and *Diceros* (Perissodactyla), had differences between the maximum and the minimum angles during a stance of between 10° and 20°, which means that $\theta_{sm-t}$ changed within ±10° from the middle-value. Two taxa, *i.e.*, *Ursus* (Carnivora), and *Ceropithecus* (Primates) had differences between the maximum angle and minimum angle during a stance of less than 30°, which means that $\theta_{sm-t}$ changed within ±15° from the middle-value. While *Macropus* (Diprotodontia) had the largest difference between the maximum and minimum angles during a stance, (31.8°), with $\theta_{sm-t}$ changing within ±16° from the middle-value, *Panthera* had the lowest SD (1.73°) while *Macropus* had the largest one (11.5°; Fig. 3 and Table 2).

Based on the differences between each SI among all target species, SI-1 had the smallest difference at 41.5°, and SI-13 had the largest difference at 54.8°. However, the smallest SD was observed for SI-4 (10.03°), while the largest statistically significant SD was for SI-13, (12.81°; Table 2). This is because the low $\theta_{sm-t}$ value for *Suricata* is considered as an outlier

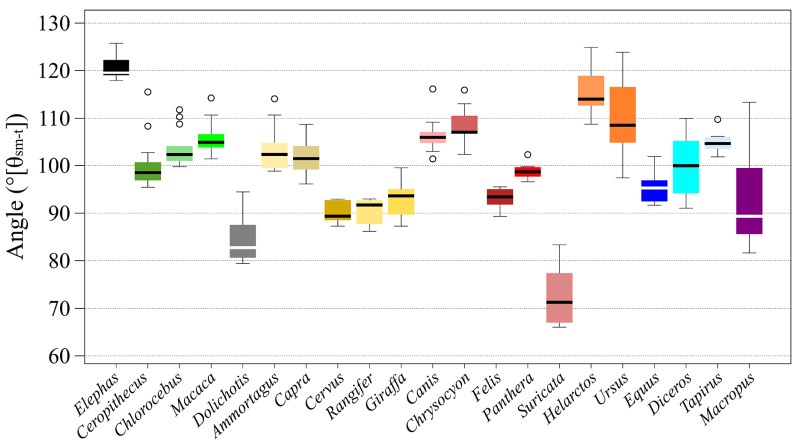

**Figure 3 The box plots of θ$_{sm−t}$ of stances of each target animals.** The x-axis shows the name of species and y-axis shows the angles in degrees. The thick bar in each box shows the median value. The top of each box shows the third quartile point, the bottom shows the first quartile point.

in SI-13 (Fig. 4). Taxonomically, Carnivora had the greatest difference between the largest and smallest angles for the same SI, being 54.8° in SI-13; this order had relatively high differences compared to the other taxa at every SI, exceeding 30° in each case (Table 2). The smallest difference was observed in Primates, being 2.9° in SI-7; this order had relatively low differences compared to the other taxa in nine out of the 13 SIs (Table 2). Based on locomotion, digitigrade species have higher difference in SI-11 (52.7°; Table 3), while digitigrade had relatively high differences in all SIs, exceeding 38° in every case. The differences for unguligrade and plantigrade fell between 11.8° and 23.3° (Table 3). Except for *Elephas* and *Macropus*, all of the examined species had positive values when θ$_{sm−t}$ of SI-2 was subtracted from SI-1, while when subtracting the values SI-2 from SI-3 values were positive for all species except *Cervus* and *Rangifer*. This indicates that these species, *Cervus*, *Rangifer*, *Elephas* and *Macropus*, started their stance phase by flexing the knee joint. The number of species with negative values increase in the subsequent steps, but the values soon became positive. The subtracted values of successive SIs were repeatedly positive and negative with in short span up to SI-9 and most species presented negative values after SI-10, showing extension of the knee joint when finishing the stance phase. The difference between successive SIs did not exceed 10° in any species, therefore, θ$_{sm−t}$ smoothly transited and changed in small amounts during a stance phase (Table 4).

According to the results of the θ$_{sm−t}$ transition analysis, every studied species had relatively small differences between maximum and minimum θ$_{sm−t}$ values during the stance phase (Figs. 3, 4, and Table 2). This showed that the total stance differences among the target animals were small; thus, θ$_{ave}$ values were representative of each species. Accordingly, we analyzed the relationships between θ$_{ave}$ and body mass. The resulting correlation coefficient (*r*) for all target animals was 0.30 with a *p*-value of 0.19 and 19 degrees of freedom (d.f.; Table 5). The correlation between the body mass and θ$_{ave}$ of each taxon was also calculated, which was significant only for Carnivora (*r* = 0.81, *p* = 0.028, d.f. = 5). The correlation between body mass and θ$_{ave}$ for each locomotor mode was only

**Table 2 Angles between the tibia shaft and *m. semimembranosus* in taxonomical order.**

| Target | SI-1 | SI-2 | SI-3 | SI-4 | SI-5 | SI-6 | SI-7 | SI-8 | SI-9 | SI-10 | SI-11 | SI-12 | SI-13 | Diff. | $\theta_{ave}$ | SD |
|---|---|---|---|---|---|---|---|---|---|---|---|---|---|---|---|---|
| Proboscidea | | | | | | | | | | | | | | | | |
| *Elephas* | 121.6 | 125.8 | 124.1 | 122.1 | 122.8 | 119.7 | 117.9 | 119.5 | 119.5 | 119.2 | 119.8 | 119.2 | 118.1 | 7.9 | 120.7 | 2.40 |
| Primates | | | | | | | | | | | | | | | | |
| *Cercopithecus* | 115.6 | 108.4 | 102.8 | 100.2 | 100.7 | 97.7 | 98.8 | 95.6 | 95.5 | 97.0 | 96.9 | 96.3 | 98.2 | 20.1 | 100.3 | 5.64 |
| *Chlorocebus* | 110.3 | 102.9 | 100.6 | 100.0 | 102.4 | 104.1 | 101.7 | 101.1 | 100.4 | 102.5 | 103.5 | 108.8 | 111.9 | 11.9 | 103.9 | 3.94 |
| *Macaca* | 114.3 | 110.8 | 108.3 | 104.9 | 103.9 | 106.7 | 101.5 | 104.3 | 102.0 | 103.8 | 105.2 | 105.7 | 104.9 | 12.8 | 105.9 | 3.52 |
| Diff. | 5.3 | 7.9 | 7.7 | 4.9 | 3.2 | 9.0 | 2.9 | 8.7 | 6.5 | 6.8 | 8.3 | 12.5 | 13.7 | | | |
| SD | 2.75 | 4.06 | 3.99 | 2.78 | 1.61 | 4.63 | 1.61 | 4.43 | 1.75 | 3.58 | 4.42 | 6.51 | 6.87 | | | |
| Rodentia | | | | | | | | | | | | | | | | |
| *Dolichotis* | 94.5 | 90.8 | 87.6 | 88.4 | 85.9 | 83.9 | 82.9 | 82.5 | 79.5 | 80.3 | 80.7 | 80.7 | 81.2 | 15.0 | 84.5 | 4.61 |
| Artiodactyla | | | | | | | | | | | | | | | | |
| *Ammotragus* | 114.2 | 110.7 | 107.8 | 104.7 | 104.8 | 102.4 | 103.4 | 100.1 | 99.6 | 99.8 | 99.0 | 99.1 | 99.5 | 15.2 | 103.5 | 4.87 |
| *Capra* | 108.8 | 101.0 | 99.5 | 104.4 | 102.2 | 99.5 | 97.0 | 100.5 | 96.3 | 101.7 | 102.0 | 105.3 | 104.1 | 12.5 | 101.7 | 3.41 |
| *Cervus* | 93.1 | 92.8 | 92.9 | 90.7 | 89.5 | 91.1 | 89.0 | 87.9 | 87.5 | 87.3 | 88.9 | 89.3 | 93.1 | 5.8 | 90.2 | 2.18 |
| *Rangifer* | 93.0 | 92.8 | 93.0 | 92.7 | 92.2 | 91.7 | 91.2 | 87.9 | 89.8 | 86.3 | 87.9 | 88.1 | 92.5 | 6.7 | 90.7 | 2.39 |
| *Giraffa* | 99.7 | 96.7 | 95.1 | 94.9 | 94.6 | 94.9 | 93.8 | 92.6 | 90.3 | 87.3 | 89.4 | 89.8 | 88.1 | 12.4 | 92.9 | 3.65 |
| Diff. | 21.2 | 17.9 | 14.9 | 14.0 | 15.3 | 11.3 | 14.4 | 12.6 | 12.1 | 15.4 | 14.1 | 17.2 | 16.0 | | | |
| SD | 9.47 | 7.44 | 6.26 | 6.63 | 6.59 | 4.92 | 5.61 | 6.23 | 5.02 | 7.57 | 6.54 | 7.55 | 6.32 | | | |
| Carnivora | | | | | | | | | | | | | | | | |
| *Canis* | 116.2 | 109.2 | 108.0 | 106.2 | 105.4 | 106.1 | 105.0 | 105.5 | 107.2 | 103.0 | 101.7 | 103.3 | 106.7 | 14.5 | 95.6 | 3.59 |
| *Chrysocyon* | 112.0 | 110.5 | 107.1 | 106.9 | 106.9 | 102.5 | 105.1 | 106.9 | 106.7 | 107.8 | 107.6 | 113.1 | 116.1 | 13.6 | 108.4 | 3.62 |
| *Felis* | 95.6 | 93.6 | 89.5 | 91.9 | 89.6 | 92.4 | 92.2 | 90.8 | 93.9 | 94.4 | 95.7 | 95.6 | 95.0 | 6.2 | 93.1 | 2.20 |
| *Panthera* | 102.4 | 99.6 | 99.1 | 99.9 | 97.9 | 97.7 | 97.8 | 98.8 | 99.2 | 98.2 | 96.6 | 98.8 | 102.5 | 5.9 | 99.1 | 1.73 |
| *Suricata* | 83.5 | 82.0 | 79.6 | 77.3 | 75.5 | 71.3 | 73.1 | 71.0 | 68.8 | 66.9 | 67.1 | 66.7 | 66.0 | 17.5 | 73.0 | 6.09 |
| *Helarctos* | 125.0 | 122.1 | 118.8 | 112.9 | 114.6 | 108.8 | 110.6 | 113.3 | 110.4 | 113.2 | 114.2 | 115.5 | 120.8 | 16.2 | 115.4 | 4.89 |
| *Ursus* | 124.0 | 120.8 | 117.1 | 116.5 | 113.3 | 110.1 | 108.6 | 108.1 | 105.1 | 104.9 | 101.9 | 100.4 | 97.5 | 26.5 | 109.9 | 8.08 |
| Diff. | 41.5 | 40.1 | 39.2 | 39.2 | 39.1 | 18.8 | 37.5 | 42.3 | 41.6 | 46.3 | 47.1 | 48.8 | 54.8 | | | |
| SD | 15.32 | 14.59 | 14.34 | 13.46 | 14.02 | 13.48 | 13.02 | 14.38 | 14.32 | 15.18 | 14.97 | 16.08 | 17.93 | | | |
| Perissodactyla | | | | | | | | | | | | | | | | |
| *Equus* | 102.8 | 99.7 | 99.5 | 96.8 | 96.2 | 95.3 | 95.6 | 95.3 | 92.5 | 91.7 | 92.0 | 92.4 | 92.5 | 11.1 | 95.6 | 3.32 |
| *Diceros* | 110.0 | 109.9 | 106.5 | 105.3 | 103.7 | 104.8 | 98.6 | 100.0 | 95.4 | 94.3 | 91.1 | 91.1 | 91.5 | 18.9 | 100.2 | 7.01 |
| *Tapirus* | 109.9 | 106.0 | 103.6 | 106.0 | 106.3 | 106.3 | 104.5 | 105.1 | 104.9 | 102.2 | 102.0 | 102.0 | 104.8 | 7.9 | 104.9 | 2.19 |
| Diff. | 7.2 | 10.2 | 7.0 | 9.2 | 10.1 | 11.0 | 8.9 | 9.8 | 12.4 | 10.5 | 10.9 | 10.9 | 13.3 | | | |
| SD | 4.55 | 5.13 | 3.51 | 5.09 | 5.22 | 5.96 | 4.51 | 4.90 | 6.45 | 5.43 | 6.03 | 5.92 | 7.38 | | | |
| Diprotodontia | | | | | | | | | | | | | | | | |
| *Macropus* | 113.4 | 113.6 | 109.0 | 99.4 | 93.2 | 85.7 | 84.1 | 81.8 | 82.1 | 87.6 | 86.4 | 92.5 | 89.4 | 31.8 | 93.7 | 11.50 |
| Total | | | | | | | | | | | | | | | | |
| Diff. | 41.5 | 43.8 | 44.5 | 44.8 | 47.3 | 48.4 | 44.8 | 48.5 | 50.7 | 52.3 | 52.7 | 52.5 | 54.8 | | | |
| Mean | 107.6 | 104.7 | 102.4 | 101.1 | 100.1 | 98.7 | 97.7 | 97.6 | 96.5 | 96.6 | 96.6 | 97.8 | 98.8 | | 99.7 | |
| SD | 11.18 | 11.18 | 10.73 | 10.03 | 10.64 | 10.61 | 10.22 | 11.33 | 11.39 | 11.66 | 11.56 | 12.05 | 12.81 | | | |

**Note:**
The angle data are the average of three stances collected in our study. The differences between the maximum and the minimum angle were calculated (Diff). To visualize the variability, the standard deviation of each row and column was also calculated (SD). The raw data are in the appendix.

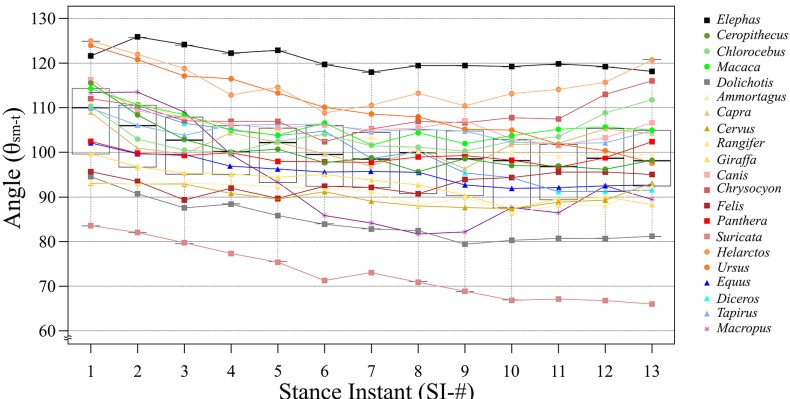

**Figure 4 The line and box charts of $\theta_{sm-t}$ transition of stances of each target animals.** The x-axis shows SIs and the y-axis shows the $\theta_{sm-t}$ in degrees. Each line and plot color are identical to the colors of each box in Fig. 2. The circle plots showed the plantigrades, the triangle plots show the unguligrade, the square plots show the digitigrades, and an asterisk show other walk style. The top and bottom bars showed the maximum angles and the smallest angles, respectively. The plots under bars are outliers.

**Table 3 Angles between the tibia shaft and *m. semimembranosus* in ambulator style order.**

| Locomotor mode | SI-1 | SI-2 | SI-3 | SI-4 | SI-5 | SI-6 | SI-7 | SI-8 | SI-9 | SI-10 | SI-11 | SI-12 | SI-13 | $\theta_{ave}$ | SD |
|---|---|---|---|---|---|---|---|---|---|---|---|---|---|---|---|
| Unguligrade | | | | | | | | | | | | | | | |
| *Ammotragus* | 114.2 | 110.7 | 107.8 | 104.7 | 104.8 | 102.4 | 103.4 | 100.1 | 99.6 | 99.8 | 99.0 | 99.1 | 99.5 | 103.5 | 4.87 |
| *Capra* | 108.8 | 101.0 | 99.5 | 104.4 | 102.2 | 99.5 | 97.0 | 100.5 | 96.3 | 101.7 | 102.0 | 105.3 | 104.1 | 101.7 | 3.41 |
| *Cervus* | 93.1 | 92.8 | 92.9 | 90.7 | 89.5 | 91.1 | 89.0 | 87.9 | 87.5 | 87.3 | 88.9 | 89.3 | 93.1 | 90.2 | 2.18 |
| *Rangifer* | 93.0 | 92.8 | 93.0 | 92.7 | 92.2 | 91.7 | 91.2 | 87.9 | 89.8 | 86.3 | 87.9 | 88.1 | 92.5 | 90.7 | 2.39 |
| *Giraffa* | 99.7 | 96.7 | 95.1 | 94.9 | 94.6 | 94.9 | 93.8 | 92.6 | 90.3 | 87.3 | 89.4 | 89.8 | 88.1 | 92.9 | 3.65 |
| *Equus* | 102.8 | 99.7 | 99.5 | 96.8 | 96.2 | 95.3 | 95.6 | 95.3 | 92.5 | 91.7 | 92.0 | 92.4 | 92.5 | 95.6 | 3.32 |
| *Diceros* | 110.0 | 109.9 | 106.5 | 105.3 | 103.7 | 104.8 | 98.6 | 100.0 | 95.4 | 94.3 | 91.1 | 91.1 | 91.5 | 100.2 | 7.01 |
| *Tapirus* | 109.9 | 106.0 | 103.6 | 106.0 | 106.3 | 106.3 | 104.5 | 105.1 | 104.9 | 102.2 | 102.0 | 102.0 | 104.8 | 104.9 | 2.19 |
| Diff. | 21.25 | 17.9 | 14.9 | 15.3 | 16.8 | 15.2 | 15.5 | 17.2 | 17.4 | 15.9 | 14.1 | 17.2 | 16.7 | | |
| SD | 8.10 | 7.08 | 5.84 | 6.31 | 6.37 | 5.85 | 5.45 | 6.31 | 5.72 | 6.68 | 5.96 | 6.55 | 6.22 | | |
| Digitigrade | | | | | | | | | | | | | | | |
| *Elephas* | 121.6 | 125.8 | 124.1 | 122.1 | 122.8 | 119.7 | 117.9 | 119.5 | 119.5 | 119.2 | 119.8 | 119.2 | 118.1 | 120.7 | 2.40 |
| *Dolichotis* | 94.5 | 90.8 | 87.6 | 88.4 | 85.9 | 83.9 | 82.9 | 82.5 | 79.5 | 80.3 | 80.7 | 80.7 | 81.2 | 84.5 | 4.61 |
| *Canis* | 116.2 | 109.2 | 108.0 | 106.2 | 105.4 | 106.1 | 105.0 | 105.5 | 107.2 | 103.0 | 101.7 | 103.3 | 106.7 | 95.6 | 3.59 |
| *Chrysocyon* | 112.0 | 110.5 | 107.1 | 106.9 | 106.9 | 102.5 | 105.1 | 106.9 | 106.7 | 107.8 | 107.6 | 113.1 | 116.1 | 108.4 | 3.62 |
| *Felis* | 95.6 | 93.6 | 89.5 | 91.9 | 89.6 | 92.4 | 92.2 | 90.8 | 93.9 | 94.4 | 95.7 | 95.6 | 95.0 | 93.1 | 2.20 |
| *Panthera* | 102.4 | 99.6 | 99.1 | 99.9 | 97.9 | 97.7 | 97.8 | 98.8 | 99.2 | 98.2 | 96.6 | 98.8 | 102.5 | 99.1 | 1.73 |
| *Suricata* | 83.5 | 82.0 | 79.6 | 77.3 | 75.5 | 71.3 | 73.1 | 71.0 | 68.8 | 66.9 | 67.1 | 66.7 | 66.0 | 73.0 | 6.09 |
| Diff. | 38.1 | 43.3 | 44.5 | 44.8 | 47.3 | 48.4 | 44.8 | 48.5 | 50.7 | 52.3 | 52.7 | 52.5 | 52.1 | | |
| SD | 13.56 | 14.69 | 15.12 | 14.63 | 15.67 | 15.68 | 15.04 | 16.33 | 17.39 | 17.44 | 17.33 | 18.18 | 18.92 | | |
| Plantigrade | | | | | | | | | | | | | | | |
| *Cercopithecus* | 115.6 | 108.4 | 102.8 | 100.2 | 100.7 | 97.7 | 98.8 | 95.6 | 95.5 | 97.0 | 96.9 | 96.3 | 98.2 | 100.3 | 5.64 |
| *Chlorocebus* | 110.3 | 102.9 | 100.6 | 100.0 | 102.4 | 104.1 | 101.7 | 101.1 | 100.4 | 102.5 | 103.5 | 108.8 | 111.9 | 103.9 | 3.94 |

| Locomotor mode | SI-1 | SI-2 | SI-3 | SI-4 | SI-5 | SI-6 | SI-7 | SI-8 | SI-9 | SI-10 | SI-11 | SI-12 | SI-13 | $\theta_{ave}$ | SD |
|---|---|---|---|---|---|---|---|---|---|---|---|---|---|---|---|
| *Macaca* | 114.3 | 110.8 | 108.3 | 104.9 | 103.9 | 106.7 | 101.5 | 104.3 | 102.0 | 103.8 | 105.2 | 105.7 | 104.9 | 105.9 | 3.52 |
| *Helarctos* | 125.0 | 122.1 | 118.8 | 112.9 | 114.6 | 108.8 | 110.6 | 113.3 | 110.4 | 113.2 | 114.2 | 115.5 | 120.8 | 115.4 | 4.89 |
| *Ursus* | 124.0 | 120.8 | 117.1 | 116.5 | 113.3 | 110.1 | 108.6 | 108.1 | 105.1 | 104.9 | 101.9 | 100.4 | 97.5 | 109.9 | 8.08 |
| Diff. | 14.7 | 19.2 | 18.2 | 16.5 | 13.9 | 12.4 | 11.8 | 17.7 | 14.9 | 16.2 | 17.3 | 19.2 | 23.3 | | |
| SD | 8.10 | 7.08 | 5.84 | 6.31 | 6.37 | 5.85 | 5.45 | 6.31 | 5.72 | 6.68 | 5.96 | 6.55 | 6.22 | | |
| Other | | | | | | | | | | | | | | | |
| *Macropus* | 113.4 | 113.6 | 109.0 | 99.4 | 93.2 | 85.7 | 84.1 | 81.8 | 82.1 | 87.6 | 86.4 | 92.5 | 89.4 | 93.7 | 11.50 |

Note:
Data for each SI angle are the same as in Table 2. The $\theta_{ave}$ are the average values of each stance (a series of SI-1 to SI-13). The differences between the maximum and the minimum angle were calculated (Diff). To visualize the variability, the standard deviation of each row and column was also calculated (SD). The raw data are in the appendix.

**Table 4 Subtracted values of each SIs: subtracted $\theta_{sm-t}$ less the previous $\theta_{sm-t}$.**

| Target | SI1–SI2 | SI2–SI3 | SI3–SI4 | SI4–SI5 | SI5–SI6 | SI6–SI7 | SI7–SI8 | SI8–SI9 | SI9–SI10 | SI10–SI11 | SI11–SI12 | SI12–SI13 | Locomotor mode |
|---|---|---|---|---|---|---|---|---|---|---|---|---|---|
| *Elephas* | *−4.25* | 1.71 | 1.97 | *−0.66* | 3.11 | 1.75 | *−1.52* | 0.00 | 0.29 | *−0.64* | 0.62 | 1.05 | D |
| *Cercopithecus* | 7.21 | 5.62 | 2.61 | *−0.52* | 2.97 | *−1.08* | 3.23 | *−2.97* | 1.50 | 0.19 | 0.53 | *−1.83* | P |
| *Chlorocebus* | 7.42 | 2.33 | 0.61 | *−2.46* | *−1.67* | 2.43 | 0.57 | 0.71 | *−2.14* | *−0.98* | *−5.32* | *−3.07* | P |
| *Macaca* | 3.49 | 2.51 | 3.44 | 0.97 | *−2.81* | 5.19 | *−2.82* | 2.30 | *−1.73* | *−1.46* | *−0.48* | 0.78 | P |
| *Dolichotis* | 3.73 | 3.20 | *−0.86* | 2.47 | 2.00 | 1.07 | 0.35 | 3.02 | *−0.81* | *−0.34* | *−0.05* | *−0.46* | D |
| *Ammotragus* | 3.48 | 2.88 | 3.07 | *−0.10* | 2.40 | *−0.94* | 3.30 | 0.51 | *−0.23* | 0.81 | *−0.19* | *−0.32* | U |
| *Capra* | 7.79 | 1.53 | *−4.93* | 2.26 | 2.66 | 2.51 | *−3.47* | 4.19 | *−5.39* | *−0.28* | *−3.33* | 1.20 | U |
| *Cervus* | 0.28 | *−0.06* | 2.19 | 1.15 | *−1.60* | 2.13 | 1.13 | 0.36 | 0.22 | *−1.57* | *−0.43* | *−3.76* | U |
| *Rangifer* | 0.19 | *−0.22* | 0.31 | 0.55 | 0.47 | 0.52 | 3.30 | *−1.95* | 3.51 | *−1.54* | *−0.19* | *−4.42* | U |
| *Giraffa* | 3.00 | 1.60 | 0.19 | 0.69 | *−0.67* | 1.14 | 1.17 | 2.30 | 2.98 | *−2.11* | *−0.37* | 1.76 | U |
| *Canis* | 6.97 | 1.29 | 1.71 | 0.80 | *−0.62* | 1.03 | *−0.46* | *−1.68* | 4.16 | 1.35 | *−1.64* | *−3.42* | D |
| *Chrysocyon* | 1.49 | 3.41 | 0.13 | *−0.02* | 4.49 | *−2.67* | *−1.8* | 0.24 | *−1.12* | 0.24 | *−5.56* | *−3.01* | D |
| *Felis* | 2.03 | 4.02 | *−2.36* | 2.32 | *−2.84* | 0.24 | 1.40 | *−3.08* | *−0.51* | *−1.29* | 0.08 | 0.55 | D |
| *Panthera* | 2.79 | 0.49 | *−0.75* | 2.02 | 0.16 | *−0.07* | *−1.00* | *−0.41* | 0.96 | 1.58 | *−2.12* | *−3.76* | D |
| *Suricata* | 1.56 | 2.41 | 2.25 | 1.85 | 4.18 | *−1.85* | 2.14 | 2.21 | 1.92 | *−0.22* | 0.44 | 0.66 | D |
| *Helarctos* | 2.91 | 3.30 | 5.90 | *−1.65* | 5.76 | *−1.80* | *−2.70* | 2.89 | *−2.87* | *−0.94* | *−1.33* | *−5.30* | P |
| *Ursus* | 3.19 | 3.66 | 0.58 | 3.24 | 3.22 | 1.46 | 0.56 | 2.92 | 0.23 | 3.02 | 1.50 | 2.87 | P |
| *Equus* | 2.36 | 0.18 | 2.73 | 0.63 | 0.88 | *−0.32* | 0.29 | 2.78 | 0.82 | *−0.31* | *−0.41* | *−0.07* | U |
| *Diceros* | 0.16 | 3.36 | 1.21 | 1.61 | *−1.15* | 6.27 | *−1.43* | 4.57 | 1.15 | 3.19 | *−0.05* | *−0.37* | U |
| *Tapirus* | 3.90 | 2.43 | *−2.37* | *−0.29* | 0.01 | 1.76 | *−0.64* | 0.27 | 2.71 | 0.17 | *−0.01* | *−2.79* | U |
| *Macropus* | *−0.20* | 4.55 | 9.63 | 6.17 | 7.47 | 1.63 | 2.27 | *−0.33* | *−5.38* | 1.14 | *−6.10* | 3.07 | O |

Note:
Each cell shows the subtracted value of $\theta_{sm-t}$. The italics cells show negative values. Locomotor mode abbreviations are the same as in Table 1. The bold cells show positive values.

significant for digitigrade ($r = 0.88$, $p = 0.01$, d.f. = 5; Table 5). Thus, there was no statistically significant correlation between $\theta_{ave}$ and body mass except for Carnivora and digitigrade. Furthermore, the $\theta_{ave}$ of all species was 99.7°, with the smallest being that of *Suricata* (73.0°), with the largest that of *Elephas* (120.7°). Therefore, more than 80% of the

**Table 5  A list of correlation coefficient (r) values and each p-value.**

| Body mass of X | Correlation coefficient | p-value (<0.05) | Degrees of freedom |
|---|---|---|---|
| All | 0.19 | 0.55 | 19 |
| Primates | 0.81 | 0.40 | 1 |
| Artiodactyla | −0.46 | 0.44 | 3 |
| Carnivora | 0.81 | 0.028 | 5 |
| Perissodactyla | 0.71 | 0.49 | 1 |
| Plantigrade | 0.82 | 0.09 | 3 |
| Digitigrade | 0.88 | 0.01 | 5 |
| Unguligrade | 0.07 | 0.87 | 6 |

**Table 6  Results of (A) ANOVA and (B) multiple comparisons.**

**A. One-way ANOVA**

| Explanatory variable | F-value | Numerator d.f. | Denominator d.f. | p-value |
|---|---|---|---|---|
| Taxa | 1.62 | 3.00 | 6.82 | 0.27 |
| Ambulatory style | 4.16 | 2.00 | 9.80 | 0.049 |
| Body mass | 0.67 | 3.00 | 6.03 | 0.60 |

**B. Multiple comparisons**

| Pair | Mean diff. | Standard error | d.f. | p-value |
|---|---|---|---|---|
| Unguligrade *vs* digitigrade | 0.44 | 6.38 | 7.40 | 0.997 |
| Unguligrade *vs* plantigrade | 9.65 | 3.29 | 8.77 | 0.041 |
| Digitigrade *vs* plantigrade | −9.21 | 6.55 | 7.97 | 0.383 |

**Notes:**
List (A) shows results of one-way ANOVA. List (B) shows results of multiple comparisons.
d.f., degree of freedom; diff., difference.

targets (17/21) had an angle between 90° and 110° (Table 2) including all Artiodactyla, Perissodactyla, and five of the seven Carnivora assessed in our study.

ANOVAs of $\theta_{ave}$ values were used to compare taxa, locomotor mode, and body mass. Only locomotor mode was statistically significant ($p = 0.049$; Table 6A). Furthermore, the multiple comparisons among locomotor modes showed a significant difference between unguligrade and plantigrade species ($p = 0.04$; Table 6B).

# DISCUSSION

Quadrupedal animals use their limbs for inverted pendulum-like movements (*Cavagna, Heglund & Taylor, 1977*; *Griffin, Main & Farley, 2004*). Physically, the swing velocity depends on the rod length; in terrestrial mammals, the swing speed affects the walking velocity. In this regard, limbs are the only structure that control the distance between the ground and body trunk; therefore, the rod length depends on the joint angles. The knee joint receives forces to flex from several influencing factors, such as the collision at touchdown, gravity, and a rising the center of mass. This means that the extensor muscles react immediately against flexion. In addition, quadrupedal mammals recovered up to 70%

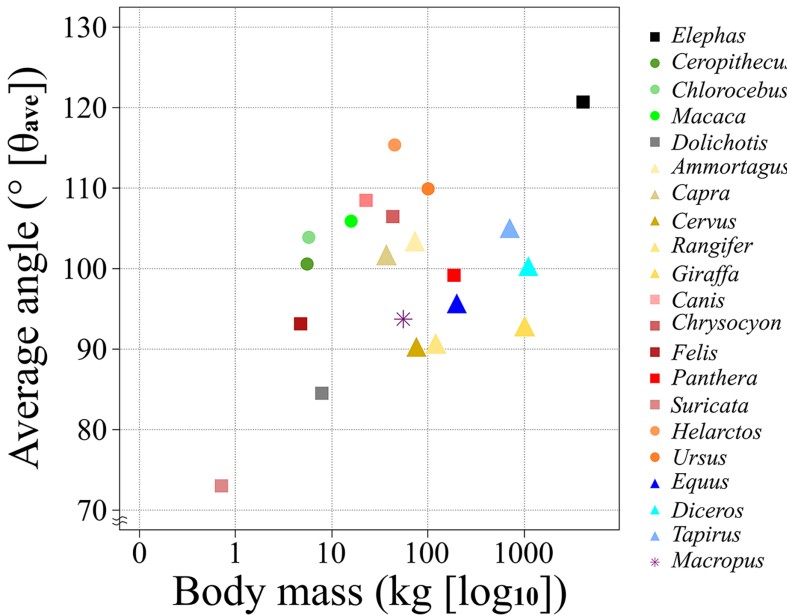

**Figure 5** **A scatter chart of the body mass and θ_ave.** The x-axis shows the body mass in the log of kg and the y-axis shows the θ_ave. Each plot color is identical to the colors of each box in Fig. 2. The circle plots showed the plantigrades, the triangle plots show the unguligrade, the square plots show the digitigrades, and an asterisk shows another walk style.

of their mechanical energy to lift and accelerate their center of mass *via* an inverted pendulum mechanism (*Griffin, Main & Farley, 2004*). Therefore, the joint angle should also be maintained constant to keep the length of the pendulum arm. Co-contraction also occurs to increase joint stiffness (*Hogan, 1984*), *i.e.*, flexor and extensor muscles are stimulated at the same time. Previous studies focused on the position of the femur while, in our study, we center the attention on the location of the ischial tuberosity of the pelvis. As the pelvis does not rotate drastically during walking, this logic also applies to $\theta_{sm-t}$. Each examined stance showed different results than our expectation, that extension and flexion periods were not completely separated as in the case of extension in the first half of a stance and flexion in the later half. The difference between $\theta_{sm-t}$ in successive SIs showed that joint flexion and extension were repeated over a short timespan (Table 4). The alternating increase and decrease in $\theta_{sm-t}$ between each SI allow quadrupedal mammals to maintain joint angles. In other words, the role of co-contraction during walking is not to fix the joint angles but to maintain the joint angles within a certain range, involving small increases and decreases in $\theta_{sm-t}$ across the broad range of studied taxa (Table 4). Therefore, the $\theta_{sm-t}$ angle transitions during one stance were small among the target species. Furthermore, the angle transition waveforms resembled among studied species (Fig. 4). *Macropus* had a unique waveform because they support their body with a tail and move both hindlimbs together (*O'Connor et al., 2014*). This walking pattern was found only in *Macropus*. Although there are different waveforms, we found that 18 out of 21 target species had only slight differences in $\theta_{sm-t}$ change (less than ±10° from the middle value) even though the largest difference was ±15.86° (Figs. 3, 4 and Table 2).

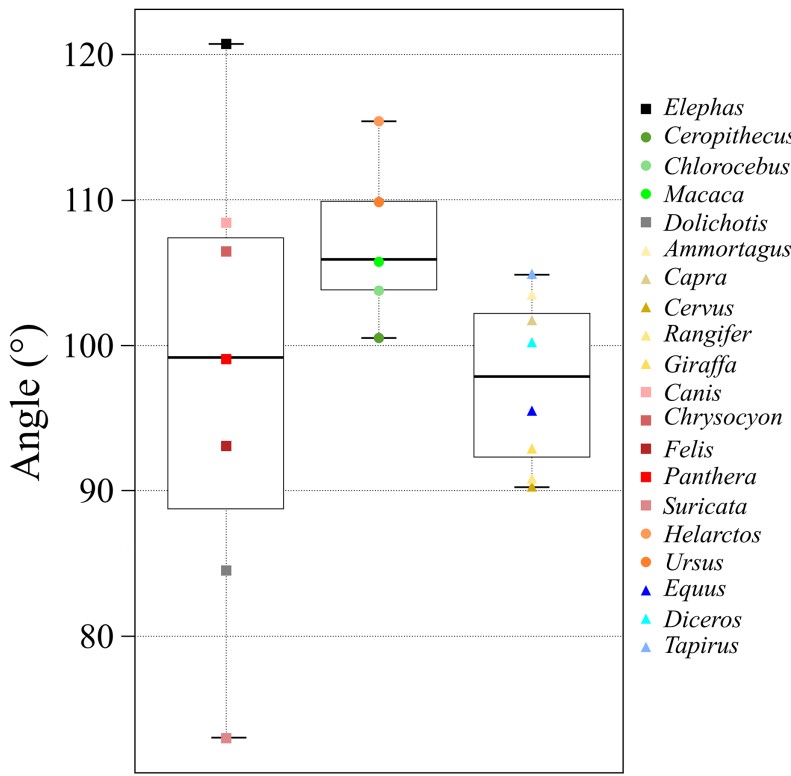

**Figure 6 A box plot of θ$_{ave}$ of each ambulatory style.** The x-axis shows the locomotor modes and the y-axis shows the θ$_{ave}$. Each plot color is identical to the colors of each box in Fig. 2. The circle plots showed the plantigrades, the triangle plots show the unguligrade, the square plots show the digitigrades.

The θ$_{ave}$ values of most of the studied species (>80%) were 100 ± 10° (*i.e.*, excluded species are *Elephas*, *Dolichotis*, *Helarctos* and *Suricata*; Table 2 and Fig. 5) including those of all three locomotor modes (*i.e.*, unguligrade, digitigrade, and plantigrade), and five out of seven orders (*i.e.*, Primates, Artiodactyla, Carnivora, Perissodactyla, and Marsupialia), with slight differences between unguligrade and plantigrade ($p = 0.04$; Table 6B and Fig. 6). Species within this range also had a wide range of the body masses, from 4.8 kg (*Felis*) to 1,100 kg (*Diceros*; Tables 1 and 3). The effective mechanical advantage (EMA) is one means of estimating mammalian limb posture; the larger EMA, the more upright the posture, with the largest species typically having greater EMA (*Biewener, 1989*, *1990*, *2005*; *Dick & Clemente, 2017*). Even when for the new measurement proposed in our work a slight correlation can be observed between the knee angle and body mass (Fig. 5), this correlation is not significant ($r = 0.3$ and $p = 0.19$, d.f. = 19), when considering all studied species. Also, our findings show that θ$_{ave}$ is much less variable than EMA. Such a difference between studies is due to the differences in angle-measurement positions. The ischial tuberosity, to which the *m. semimembranosus* is attached, is located near the posterior end of the pelvis. The horizontal or vertical orientation of the pelvis is related to body mass, with a larger body mass having a more upright orientation (*Polly, 2007*). Therefore, a larger

body mass has a larger difference between the angle of the femur-tibia (the traditional knee joint angle, as previously standardized) than the *m. semimembranosus*-tibia ($\theta_{sm-t}$ and $\theta_{ave}$, as proposed in our study). In other words, $\theta_{em-t}$ and $\theta_{ave}$ have the advantage of reflecting the small differences between these angles in large-body-mass species and small body mass species. Furthermore, the EMA does not increase linearly above species weighing 300 kg (*Biewener, 1990*, *2005*; *Dick & Clemente, 2017*), and felids have a crouched posture even with a larger body mass (*Day & Jayne, 2007*; *Dick & Clemente, 2017*). In contrast, $\theta_{ave}$ shows constant values ($100 \pm 10°$) among all locomotor modes and a wide body mass range (4.5–1,100 kg; Tables 1 and 3).

In addition, we measured $\theta_{sm-t}$ based on three points on the skeleton the ischial tuberosity interior-proximal end of the tibia, and distal end of the tibia (Fig. 1). This indicates that the position of the ischial tuberosity and tibia can be fixed with $100 \pm 10°$ on extant terrestrial quadrupedal mammals, including those with no closely related extant descendants. If a femur exists or its shape can be estimated, the limb posture can be reconstructed with higher accuracy using our approach because both the caput femoris and the distal end of the femur can be placed in the determined positions, which are the acetabulum and the proximal end of the tibia, respectively. For example, Desmostylia has been previously reconstructed in several different postures even whole-body skeletons exist (*Shikama, 1966*; *Inuzuka, 1988*; *Domning, 2002*; *Inuzuka, Sawamura & Watabe, 2006*). Because this extinct mammal has no closely related descendants and has an extremely unusual tibia, the distal half of the tibia is strongly medially twisted by approximately about 40° (*Shikama, 1966*; *Inuzuka, 1988*), no extant mammals have tibias resembling to those of Desmostylia. The $\theta_{ave}$ value, which is $100 \pm 10°$, is not strongly affected by taxonomy, body mass, and locomotor mode, and therefore, this degree can be applied to Desmostylia.

## CONCLUSION

Stimulation of the agonist and antagonist muscles, known as co-contraction, increases joint stiffness. In the case of the knee joint angle, our result show that $\theta_{sm-t}$ transition shown as almost flat wave-form; $\theta_{sm-t}$ did not change drastically during the first 75% of SIs during the stance phase, and the co-contraction associated with by part of the *m. quadriceps femoris* and the *m. semimembranosus* seems to effectively supports the constant posture of the knee joint in most terrestrial mammals. More than 80% of the target animals in our study had similar $\theta_{ave}$ ($100 \pm 10°$) including species across a wide range of taxa, body masses, and locomotor modes, $\theta_{sm-t}$ is measured from three points on the skeletons. Our findings indicate that $\theta_{ave}$ can be a useful criteria for reconstructing the joint angles and posture of extinct mammals even if they have no closely related extant descendants.

The correlation between body mass and $\theta_{ave}$ by taxon, and the angles unique to taxon and locomotor modes suggest the possibility of applying a correction to $100 \pm 10°$ that could be applied to the all mammals. However, because our study focused on examining trends across a wide range of taxa, the sample size for each taxon was small. Further data collection and validation are required to obtain more accurate values for such corrections. In particular, our results showed a significant difference between unguligrade and plantigrade, therefore, it would apply more accurate correction if increase data.

In addition, we found that *Suricata* had two unique features: six out of the 13 $\theta_{sm-t}$ values were outliers when compared with the other species (Fig. 4), and the difference between $\theta_{ave}$ and *Dolichotis*, which was the second smallest species in this study, was more than 10° (Table 2 and Fig. 5). Furthermore, the difference between *Suricata* and the next smallest species of Carnivora, *Felis*, was seven-fold in terms of body mass and 20° in terms of $\theta_{ave}$. However, the difference between *Felis* and the largest species of Carnivora, *Ursus*, was greater than 20-fold in terms of body mass but less than 20° in terms of $\theta_{ave}$ (Tables 1, 2 and Fig. 5). Therefore, it is possible that the *Suricata* data affected the *r* and resulting *p*-value for Carnivora. This is probably because *Suricata* spends a lot of time underground, which limits the required height to lift the trunk and limbs of the body. As such, further data from subterranean species are necessary to confirm this hypothesis.

# ACKNOWLEDGEMENTS

We thank Higashi Park Zoological Gardens, Higashiyama Zoo and Botanical Garden, Hitachi Kaminé Zoo, Toyohashi Zoo and Botanical Park, and Ueno Zoological Garden for permission to record the animals under their care. We also thank Yoko Tajima and Shin-ichiro Kawada (National Museum of Nature and Science) for providing access to the extant mammal collections under their care, and Naomi Wada (Yamaguchi University) for supplying the camera equipment. We thank Katsuo Sashida (University of Tsukuba, now Mahidol University, Thailand), Sachiko Agematsu (University of Tsukuba), Kohei Tanaka (University of Tsukuba), Ikuko Tanaka (University of Tsukuba, now Geological Survey of Japan) and Yasunari Shigeta (NMNS/University of Tsukuba) for providing helpful advice, discussion, and generous encouragement during the course of our study. We would like to thank Editage for English language editing. Finally, we greatly appreciate the editor and reviewers for their helpful suggestions to improve this article.

## Funding

The authors received no funding for this work.

## Competing Interests

The authors declare that they have no competing interests.

## Author Contributions

- Fumihiro Mizuno conceived and designed the experiments, performed the experiments, analyzed the data, prepared figures and/or tables, authored or reviewed drafts of the article, and approved the final draft.
- Naoki Kohno conceived and designed the experiments, authored or reviewed drafts of the article, and approved the final draft.

## Data Availability

The raw measurements are available in the Supplemental File.

## Supplemental Information

Supplemental information for this article can be found online at http://dx.doi.org/10.7717/peerj.15379#supplemental-information.

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
