# Peer review of "New genicular joint angle criteria for flexor muscle (Musculus Semimembranosus) during the terrestrial mammals walking"

_PeerJ, doi:10.7717/peerj.15379_

## Round 0.1 · original submission · Major Revisions

I have now received two rather discordant reviews of your manuscript. In spite of this, both reviewers provided several points that should be addressed in order to improve the manuscript. Reviewer #1 is not convinced of the advantage of the measures that you present in your manuscript in relation to the classical measurement. A stronger explanation should be provided in relation to this point. The way you obtained your data is also not clear enough. Reviewer #1 suggests a number of valuable ideas to improve your work. Reviewer #2 asks for more details about your data and methodology. Thus, both reviewers present concerns about your methodology, and I find their objections highly relevant. For the revised version of your work, I would like you to perform the methodological modifications and clarification suggested by our reviewers. This point is mandatory as I am not convinced of your conclusions owed to the lack of clarity about the data and their analyses. Additional recommendations of both reviewers are mostly constructive, and attention to their suggestions will serve to improve the manuscript.

·

Excellent Review

This review has been rated excellent by staff (in the top 15% of reviews)
EDITOR COMMENT
It is a very constructive and detailed review. Several interesting publications are suggested to be taken into account. Alternatives are proposed for some methodological details and even writing. Many thanks to the reviewer for her effort.

Basic reporting

The paper entitled “A flexor muscle (musculus semimembranosus) shows the genicular joint angle when terrestrial mammals walk” (#76662) is about contrasting angles of joints of different mammal hind limbs while walking, to search for a common (or not) pattern in angle of the hind limb among mammals with different size, phylogenetic relationships and locomotor modes (including limb posture and gait). The paper is concise and respects the journal´s general structure, but it has lots of grammatical errors. I am so sorry to say that I believe that language has been a very large problem for writing ideas, which are not always clear, or at least, I am not sure to understand; please, take this into account when reading my comments, as I could potentially misunderstood what you tried to say or may be the grammar is not correct and the sentence say something different to what you intended. As nonnative English speaker I really sympathize with the authors, but clarity in language is really important. Tables are in general important, but lacking a few acronym descriptions (see below) and figures can be improved by adding some details (see specific comments)

A few comments on writing:
- Please add the definition of the stance phase of the gait cycle for non-specialist readers.
- Along the paper you use the word stance alone, but you probably should use stance phase, as this is a part of the gait cycle, and the word stance only does not reflect that.

Specific comments
Fig. 1. It would be nice if you could have a picture with better resolution for the measurement in A.
Fig. 3 and 4. Please add color coding in each figure if possible.
Table 2. what does Diff stands for?

Experimental design

I have one major concern, and it is why would the authors choose this way of measuring the angle of the knee joint, what advantage does it have in relation to the classical measurement? Why, if there are so many data using the measurement of the knee joint angle using the hip, knee and ankle joints should we change to measure this new angle? What is the difference? Although presenting a slightly different approach, i.e., measuring angle´s “knee joint” with the line of action of the muscle semimembranosus, it is not clear to me why they choose to select this variation. Even more, the authors claim that semimembranosus is not very different from the femur position itself, so I do not see any other reason. Maybe it is easier to measure because hip joint is a little bit hard to find in videos, but even though, it has been done extensively before. Is this new measurement comparable to the classical one?
This concern obviously leads to a second part which is, you should compare both measurements to see if it gives different or similar results, and if they are different, what would that mean?

Additionally, in the Methods section I have some questions.
Line 124: How did you decided the intervals? I imagine you divided the time that each species took for completing the 75% of the swing phase and divided it in 13? Please explain this to make clear that time as such was not a factor included in the analysis. I understand that you did this to make every instant steppe comparable in every species, please clarify this in the text.
Did you use a fixed tripod or the camara was moving while filming? How far was the camera to the animals when filming? Did you consider this for all species? As it may alter the angle of perception and thus, measurements.
130-131: I am sorry, what do you mean when you say each animal has three stances? I think you referee to the three steps for each filming shot and one measurement per picture, that makes thirteen, so, in that case I suggest rephrasing, let me suggest something like: For each specimen, we measured this angle in each of the thirteen photographs described above.
Did you take a single filming shot for each animal? Or did you have repetitions? Did you choose the best filming shot among several? Please clarify this in the text.
It would be also important, because you´ve worked with zoo specimens to declare that none of them present visible or known malformations, as a lot of skeleton specimens from museums are known to diverge morphologically from wild specimens.

I also have some concerns in how some things are described, things I believe have to do with concepts, content and idea developments so I mention them in this section of experiment design:
Title
"A flexor muscle (musculus semimembranosus) shows the genicular joint angle when terrestrial mammals walk"
The muscle does not show the angle of the joint, you are using the primary line of action of the muscle to measure the knee joint angle. I understand that you are using the primary line of action of semimembranosus to try to find a pattern in the relative position of long bones of hind limbs among different mammals. I suggest adjusting the title to better reflect your objectives. Additionally, the muscle semimembranosus has many different structures throughout mammals, it would be nice if you acknowledge that, and point that what you are using to measure is the line of action of the belly of the muscle that it is always present. I have not checked the structure and attachments in every considered species in this work, but it is important to know that authors did check every species, together with the cites they used to check on the muscle descriptions.

Abstract
Line 14-15: The angles of the knee joints are NOT constant while walking or moving, that’s how legs detach from the ground and complete the cycle, they are constant while standing and they change among mammalian species.

Introduction
Line 64: Written like this it sounds like knees are responsible of controlling limb motion completely, thus a little more context is needed. Pandy et al says that “the knee … function primarily as damping mechanisms, absorbing energy, and controlling limb motion. This latter result correlates well with experimental observations of human walking, where the knee joint is reportedly responsible for controlling overall motion of the body's center of gravity during stance phase”. Please check to which of the two roles of motion you mean, displacement of center of gravity or damping mechanism.
Line 71: I think adding the term agonistic generates confusion, please leave antagonistic only, as the term antagonistic summarizes that both muscles make opposite movements, no matter the direction of the resulting movement.
204-205: “In other words, the knee joint angle is mainly depended on the angle between the m. semimembranosus and the tibia.” I do not think the knee joint angle is “dependent” on the angle between the semimembranosus and the tibia, they can be associated, but implications are very different in both cases. The knee joint angle depends on a lot of other things, e.g. articular surfaces of tibia and femur, menisci, tension of other muscles (semitendinosus for example which has a very similar structure to semimembranosus and attaching more distally it can affect knee flexion even more), not only the degree of tension of the m. semimembranosus.

Discussion
No comments are made about other issues related to reconstructing a fossils posture, like articular surfaces, size, locomotion. Limb postures in extant mammals have been associated to several aspects, e.g., “Limb posture is an important attribute of animal body plans because it influences the patterns of movements and muscle activity that can contribute to propulsion, and it affects the loading of the appendicular bones (reviewed in Blob, 2001; Biewener, 2005). Whether animals are standing or moving, limb postures in which the joints are more closely aligned vertically with the point of limb attachment are likely to reduce both the muscular effort required to prevent limb collapse and the bending forces as a result of increasing the portion of the load born by simple compression of the long bones of the limb. This theoretical advantage of a straighter upright limb has long been used in part to explain some gross trends in limb postures including the evolutionary transition from a ‘sprawling’ to an ‘erect’ posture, differences between cursorial and non- cursorial mammals and differences associated with increased animal size (Osborn, 1900; Gregory, 1912; Howell, 1944; Gray, 1968; Bakker, 1971; Jenkins, 1971; Charig, 1972; Gambaryan, 1974; McMahon, 1975; Alexander, 1977; Biewener, 1983a)…, “the generalization that limb erectness increases with increased size is supported by some size-dependent scaling relationships of appendicular skeletal anatomy as well as a limited number of direct observations of limb posture during locomotion” (Day and Jane, 2007).

I would like to suggest a few ideas to the authors to consider, which I believe can improve the significance of their findings.
1) a good way to position the measurement you propose would be to compare results with the classical measurement. I did not perform a deep search, but I am sure you will find data on knee angles in mammals (Gambaryan 1974 has several, even some of the same species), so it should be easy to compare.
2) You could also try to perform non parametric analysis to compare statistical differences among groups sharing different postures (i.e., plantigrade, digitigrade and ungulate), or by size, or taxonomically. This would be really interesting.
3) You could also try the correlation of size within groups. I think you do not have a good correlation because you are not separating plantigrade, digitigrade and ungulates, if you so, I think it will correlate positively.
4) Also, I was not able to find papers suggesting results similar to yours, but such a generalized pattern in mammals should be noteworthy. Please report previous findings associated to knee angles and discuss them (e.g., those from Gambaryan 1974).

Validity of the findings

I think conclusions are not exactly according to findings. On one hand, a variation of 40 degrees in the knee joint sounds a lot to claim that it is constant among “standard” mammals (who are not defined either). Maybe it is enough to focus on the 80% of studied animals which had a 20 degrees variation.
Also, the authors claim that their methodology could help reconstruct hind limb position in extinct species with no extant relatives, they actually say “having only tibia and ischium”. Lines 232-235: “This indicates that the posture of the hind limbs of terrestrial mammals are able to reconstruct with only ischium and tibia”. I find this claim entirely false if written in this way, I would say that you may reconstruct the angle considering the hip and the tibia position in a complete skeleton, or at least with femur and acetabulum, but not with only the two elements mentioned. I do think it helps in giving a general idea on how the angle measured varies among terrestrial mammals, and that, if analyzed fossils resemble in structure to some of the analyzed mammals in this paper, it can be expected for the angle to be similar or to variate in a similar manner.
Lines 232-235: “… and thus the hind limbs of extinct mammals who does not have phylogenetically closely related extant descendants are also able to reconstruct with higher accuracy.” I would be a little less confident in this conclusion, it is true that you give a lot of evidence to better reconstruct knee joints angles, but general position of the limbs depend on a lot more than the aspect discussed here; I would change it to “and thus, this new approximation to understanding hind limb postures could be applied to the study of the hind limbs of extinct mammals with no closely related extant descendants.”

Additional comments

There are important cites missing: the most important: Gambaryan. 1974. How mammals run, where you can find a very extensive analysis of a lot of groups of mammals analyzing a broad number of areas in morphology and locomotion, including joint angles.
Other papers that should be considered to check are
- Hildebrand 1985. "Walking and Running. Functional Vertebrate Morphology," eds. M. Hildebrand, D. M. Bramble, K. F. Liem, and D. B. Wake, Harvard University Press, 1985, pp. 38-57.
- Lisa M. Day and Bruce C. Jayne. 2007. Interspecific scaling of the morphology and posture of the limbs during the locomotion of cats (Felidae). The Journal of Experimental Biology 210, 642-654.

Reviewer 2 ·

Basic reporting

no comment

Experimental design

no comment

Validity of the findings

no comment

Additional comments

comments to the authors
In this manuscript the authors comparing of the knee joint angle of terrestrial mammals during their stance phase in a broad sample of taxa. This study is also particularly important because it allows reconstructing the stance and gait of extinct animals. I consider the present manuscript of interest and this studywill likely be of use to a broad spectrum of scientists.
However, there are a number of concerns which must be addressed before the manuscript can be further considered.
The introduction does a good job introducing topic of the work, however, some aspects should be clarified in the objectives (see comments below). The materials and methods are unclear. The authors should make an effort to better explain how the data were collected and the images processed. I think a graph would be helpful. Some additional information are needed in the discussion to improve the interpretation of this section. The figures need some modifications.

My comments are summarized below

Introduction
Line 48-49.- It would be interesting for the authors to give examples of knee joint angles for the different species.
Line 53-55.- The authors here may also indicate that due to our relatively limited knowledge of the relationship between skeletal morphology, limb posture and body mass.
Line 56.- … between the joint angle and skeletal morphology
It would be interesting if the authors in this paragraph indicate whether there is previous work on these aspects in other groups of mammals.
Line 92.- Indicate in relation to what... with the different gait patterns, locomotor function, posture, movement, etc.
Materials and methods
Line 96-105.- Species are listed in the table. Please delete species from the main text of the manuscript.
Line 114.- Is the camera attached to a mount? how far? Please explain.
Line 130.- The 13 angles are SI1-SI13? What program was used to measure the angles?
I think this sentence should be in the next paragraph.
Line 132-136.-It would be helpful to readers if the authors add a figure or graph that illustrates how the measurements were recorded. As it is written it is difficult to understand.
Line 133.- Delete the point.
Line 136.- Here it would be between gait pattern between species?. I don't understand.
Results
There are standard deviation values that the authors cite but are not in Table 2. For example, Line 173-174: However, the smallest standard deviation was at SI-2, 11.06, the biggest standard deviation was at SI-13, 18.29
Discussion
It would be interesting for the authors to discuss from a functional point of view how the knee joint angle, if any, is related to gait or posture types.
Line 205-206.- The authors should mention about table 3 in the results and explain it better in the discussion.
Line 207.- SI6-7? O SI1-S7?
Line 225-226.- And what style of walking would it have?..you mean digger?
Line 232-235.- How would the reconstruction be done? It would be interesting if the authors could explain this point a little more.

Figures
Fig.1.- I think the authors should replace Figure 1A with a more schematic drawing for a better understanding of the readers. The authors should do a better editing of the photo in Figure 1B or make a schematic drawing.
Fig.2-3-4.- I think the authors should also indicate the names on the x and y axes. For example in figure 2 the axis Y put angles

---

## Round 0.2 · Major Revisions

As you will see, our first reviewer suggested a number of additional changes to be considered to improve your work. I checked whether you carefully consider the suggestions of our second reviewer and I am reasonably satisfied. I note, however, that too many paragraphs are not clear enough due to language problems. I also consider that the M&M section is not sufficiently explained, nor do I understand why you didn't consider the Suricata. Your study is quite ambitious indeed, so, I agree with our first reviewer in that the taxon sampling should be as extensive as possible. I need clarity in the explanation of the M&M section, please take all the new suggestions very carefully.

·

Basic reporting

I found a greatly improved manuscript. I would like to thank the authors for their kind and complete answers, but specially for considering and including my suggestions.
It is far clearer the difference between the classical measurement of the knee joint angle and the new one presented in this work. I would only suggest that you present the theory developed in the discussion in the introduction, so you can present your prediction of a constant angle properly and leave in the discussion the different results between previous studies and yours.
In the introduction and abstract, authors referred to co-contraction as the factor increasing possibility of constant angle of the knee joint. Later in discussion, they present an energetic approach. I find the energetic point of view much more important than co-contraction for supporting your hypothesis of expecting a constant angle, I suggest you emphasize that point of view over co-contraction and please add it in the introduction section as well. At least for me, it is like the energetics is the why and co-contraction is more like the how.
Unfortunately, even when the methodology section of the first analyses is now clear, there are still some problems with methodology of the new analyses included. Please see Experimental design section below.

Also, for functional discussion I would like to offer the authors a pdf copy of Gambaryan´s book, and, as I cannot offer Hildebrand´s book, I suggest at least the following papers.
HILDEBRAND.1984. ROTATIONS OF THE LEG SEGMENTS OF THREE FAST-RUNNING CURSORS AND AN ELEPHANT. J. Mamm., 65(4):718-720.
Hildebrand & Huxley. 1985. Energy of the Oscillating legs of a fast-moving cheetah, pronghorn, jackrabbit and elephant. J. Morphology 184:23-31.

Experimental design

I have two major concerns. The first one, why did the authors exclude Suricata from the 21 species included in the first version of the manuscript? Results change substantially, and this species was the most “problematic” in the previous version because of its different “underground abilities”. Was that because of any measurement error or because it has a different locomotor mode?
If it is the first case, there is nothing to say, but if it is the second, I do not agree with this decision. In this kind of macroecological analysis, deviations of the general patterns are as valuebles as the patterns, so we need to try to see as much as the variation as possible, especially if you are trying to make such a large generalization among terrestrial mammals and to apply it to fossil species (even more, with no known extant relatives), i.e., we are inferring their modes of life. So, if measurements form Suricata are ok, I suggest that authors report results with all 21 species and as a complement, these new results excluding this singular species indicating its particularities that may make it so distant from the rest. This would be useful if extinct species studied with this new method is also deviant of the “general pattern” reported here.

The second major concern is about the new analysis presented. I am really glad that the authors have chosen to consider my previous suggestion. But there are two problems. First, in the paper it is not clear how authors performed the analysis.
In Materials and Methods they say: “Our study compared the transition of θsm-t in a stance among species, ambulatory styles (unguligrade, digitigrade, and plantigrade) and the average of the θsm-t (i.e., θave) versus body mass.”
But, in the results sections they say: “There were no variables that showed significant differences in correlation with body mass, either taxonomically or in ambulatory (Table 5).”.
In the rebuttal letter is clearer: “Response: The results of the additional work suggested by the reviewer showed there were no significant differences between the θave and the body mass of each taxon and ambulatory style (Table 5 and on Results LINE 270). Therefore, it cannot say θave correlates to the body mass.”
The second point, but much importantly, the analyses added by authors do not respond to the question intended (sorry if I wasn´t clear enough in my previous review), i.e., is the angle of the knee joint different if you compare taxonomic groups or among functional groups? You could try a simple ANOVA analysis (parametric or non-parametric, or a pairwise comparison tests according to your data) to test whether the average/range/distribution of the angle for each evaluated group (taxonomic/functional) is different to the other groups.
I also suggested authors could try to contrast the average angle of the knee joint considering the size of species. For this I suggest categorizing the studied species in discrete size categories using weight information you have (e.g., small, medium, and large) and perform an ANOVA (or selected method) to test if the angle is significantly different between size categories.
Thus, results regarding these contrasts are still lacking.

Validity of the findings

Considering observations made in Experimental design section, I would say that a relatively general pattern can be observed, even including Suricata. Thus, in a descriptive point of view, authors could claim that the general pattern SEEMS independent of taxa, body mass, and most ambulatory styles, especially if only species with now underground behavior are included; but, as Suricata deviates from the pattern, then, probably more species with underground locomotor abilities should be studied in the future. Even though, this asseveration is not statistically supported yet, it will depend on the results of contrasts suggested in the Experimental design section, if the authors choose to perform them.

Additional comments

English has been very much improved, but it is not great yet, so, with the aid to help, I took the liberty to suggest several “major” changes in writing to improve the transmission of ideas, I hope you find them helpful but if there is anything I could misunderstand or misinterpret, or you just do not like it, please feel free to ignore them and of course modify them to improve your work even more of course, as I am not an english native speaker either.

Abstract:
Authors wrote: Therefore, the angle between the m. semimembranosus and the tibia would be kept constant because of the generation of co-contraction, and consequently the constant joint angles are estimated from this muscle.
Co-contraction occurs almost all the time in most joints, even when they are moving, it does not NECESSARILY mean that all angle joints are constant because of it. I suggest rephrasing as:
Therefore, the angle between the m. semimembranosus and the tibia is expected to remain constant because of the generation of co-contraction, and consequently, in this study, the joint angles are estimated from this muscle.


Page 5, lines 121-123: When a joint angle is locked against the force to change the angle via gravity, muscles work not only the agonist muscle but also the antagonist muscle.
I am not sure I understand, but if I am correct, I would like to suggest the following: When a joint is locked to maintain a certain angle, both, agonist and antagonist muscles must work together against the gravity force.
Although for me, using agonistic gives the idea of movement, as agonistic is defined according to in which direction the part of the body in study actually moves, so quadriceps will be agonistic if I am extending me knee or antagonistic if I am flexing my knee… thus, you could also say: "both antagonistic muscles work together"

Page 5, lines 127-128: I do not completely agree with the definition given for stance phase, I am not sure whose author´s definition is that, but the most common one is: “when the foot under consideration is in contact with the floor”. I suggest the following change to include the concept I think the authors are trying to outstand:
… time during the stance phase, which is the period in which the foot under consideration is in contact with the floor and during which the limb will support the body mass.

Page 7, lines 186-187: The sill images of each this period divided so that 13 pictures including the first and the last (Fig. 2A).
This is grammatically incorrect, and I think it is not completely clear yet the division of frames, so, based on your comments in the rebuttal letter I suggest the following:
The first 75% of the stance phase of each step of every specimen was divided in 12 equal time periods (particular for each step) in order to obtain 13 pictures, including the first and last frames.

Figures 3 and 4: Would it be possible to zoom in the graphics? In Fig. 3 you could try to use 100+-25 in the y axes, so lines won´t be so overlapped and you can more easily distinguish colors. In Figure 4, both axis can be adjusted from 50-150 and 2-8.

Supp. Mat. still has the same 21 species from the first version.

---

## Round 0.3 · Minor Revisions

As you can see, the suggested revisions are still significant, but I agree with our reviewer that they can be relatively easily performed. Our reviewer did an excellent job of helping you improve your work, and I hope you will follow through with implementing these final changes.

·

Basic reporting

The paper has greatly improved from the last version. Thank you very much to the authors for considering all the suggestions and again, for your kind answers.

I have only a few more comments, three important ones, but I think they can very easily be solved.
1) I am so glad about the new analysis, but, because you did find significant differences of the 0ave associated to locomotion and taxonomical size-related variation, then your conclusions´ asseverations about independence of the 0ave measurement (regarding size, locomotion and taxonomy) MUST be softened throughout the manuscript. Actually, the modified figures allowed me to see some new details that I think they should be mentioned. For example, the very impressive deviation of Macropus from the general pattern (Fig. 4), there is nothing discussed about it, just that because it is the only species with this different type of locomotion it is not considered for analysis… I understand it is not included in the statistical analysis, but, in a general discussion this should be considered and explained (or at least try to propose an explanation). Also, Fig 4 now allows observing that Suricata and Ursus “never” start flexing the knee joint until SI13, have you think why this is like that? Also in Fig. 5, segregation between plantigrade and unguligrade species is noteworthy.
2) I think there is a miscalculation of %, Diceros angle difference is less than 20° thus, you have 18 species with less than 20°, i.e., 85% of species.
3) In material and methods, it should be included the definitions and categories you used for size, taxa and locomotion.

A few more simple comments
- English can still be improved.
- Sorry I didn´t note this before, I would not use “ambulatory” styles, I understand the correct English term would be locomotor mode, because actually, ambulatory is a locomotor mode in itself.
Figures 4 and 5 are so much better in this new size. Now that I can see better Fig. 4, It is impressive how Macropus deviates from the “general trend” (see above)

Figure 3, 5 and 6. Axis are described “upside-down”, on the x axis are the names and on the y axis are the angles.

Please, consider applying a zoom in Fig 3, cutting the beginning of the y axis (as you did in fig 6 ) and subdividing this axis in smaller parts, every ten or 20 degrees may be? All the blank space makes no sense and real differences will be more easily seen.

Fig 6. Please erase Macropus from the locomotion mode as it is not included in the boxplot.

Table 1. Ambulatory Style O is not described in the legend.

Table 2 … the standard deviation of a low and column… I guess it´s of row and column. Also… The row data are in the supplementary file, should be the raw data. These two comments should be considered also for Table 3.

Within the table, Suricata´s data are written in smaller letters. Likewise, Diff of Helarctos. The same happens in table 3 for Suricata.

Table 4 The subtracted values of each SIs: subtracted behind 0sm-t from before 0sm-t.
Please change to: Subtracted values of each SIs: subtracted 0sm-t less the previous 0sm-t.
Each cell shows the subtracted value of 0sm-t. The colored cells show negative values.
Here you say colored cells, it should say cells in italics

Experimental design

-

Validity of the findings

See general comments

---

## Round 0.4 · accepted · Accept

I am glad to accept your work. It has been a long journey to get here, but thanks to the effort of our reviewer and yours, your initial presentation has improved significantly. Congratulations.